# Neuropeptide Y in Spotted Scat (*Scatophagus Argus*), Characterization and Functional Analysis towards Feed Intake Regulation

Daniel Assan [1,2] , Yaorong Wang [3] , Umar Farouk Mustapha [1] , Charles Brighton Ndandala [1], Zhiyuan Li [1], Guang-Li Li [1] and Huapu Chen [1,2,3,*]

1   Guangdong Research Center on Reproductive Control and Breeding Technology of Indigenous Valuable Fish Species, Guangdong Provincial Key Laboratory of Pathogenic Biology and Epidemiology for Aquatic Economic Animals, Guangdong Province Famous Fish Reproduction and Breeding Engineering Technology Research Center, Fisheries College, Guangdong Ocean University, Zhanjiang 524088, China; danielassan95@gmail.com (D.A.); umarfk.gh@gmail.com (U.F.M.); charlesndandala@gmail.com (C.B.N.); g_yl903@163.com (Z.L.); guangligdou@163.com (G.-L.L.)
2   Southern Marine Science and Engineering Guangdong Laboratory, Zhanjiang 524025, China
3   Guangdong Havwii Agriculture Group Co., Ltd., Zhanjiang 524266, China; yaorongwang217@126.com
*   Correspondence: chenhp@gdou.edu.cn; Tel.: +86-18820706692; Fax: +86-759-2382459

**Abstract:** Neuropeptide Y (Npy) is an intricate neuropeptide regulating numerous physiological processes. It is a highly conserved peptide known to improve feed intake in many vertebrates, including fishes. To enlighten the mechanism of Npy in spotted scat feed intake control, we cloned and identified the Npy cDNA sequence. We further examined its expression in some tissues and explored its expression effects at different time frames (hours and days). Here, we discovered that spotted scat Npy comprised a 300 bp open reading frame (ORF) and a 99 amino acid sequence. *Npy* was identified to be expressed in all tissues examined. Using in situ hybridization examination, we proved that *npy* has a wide expression in the brain of the spotted scat. Furthermore, the expression of *npy* in the hypothalamus significantly increased one hour after feeding ($p < 0.05$). Further, it was revealed that *npy* expression significantly increased in fish that were fasted for up to 5 days and significantly increased after refeeding from the 8th to the 10th day. This suggests that Npy is an orexigenic peptide, and hence, it increases food intake and growth in the spotted scat. Additionally, results from in vitro and in vivo experiments revealed that Npy locally interacts with other appetite-regulating peptides in the spotted scat hypothalamus. This research aimed to set a fundamental study in developing the feed intake regulation, improving growth and reproduction, which is significant to the aquaculture industry of the spotted scat.

**Keywords:** cloning; fasting and refeeding; mRNA expression; *npy*; *Scatophagus argus*

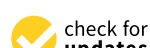



## 1. Introduction

Spotted scat; *Scatophagus argus*, a member of the family "Scatophagidae" in the order "Perciformes" is a common aquarium fish species around the world due to its colorful appearance, hardiness, slow growth, and calm behavior [1–6]. It has the Indo Pacific, the Malay Archipelago, Philippines, China, Australia, and South and Southeast Asia, especially in India and Sri Lanka, as its native range of existence [5–7]. The spotted scat is broadly distributed in freshwater, brackish water, and marine habitats in the lower reaches of rivers, estuaries, mangrove swamps, surf zone of beaches, coastal mudflats, and harbors [6,8–10]. It has a high demand in aquarium fish markets and has been reported to fetch a good market price, and due to its good nutrient quality and taste, it is popular as a food fish in the South and Southeast Asian countries [3,5,9,11,12].

Presently, numerous appetite-regulating peptides have been identified in several fish species, including neuropeptide Y (Npy) [13–15]. Neuropeptide Y (Npy), an extremely

conserved neuroendocrine peptide [16,17], is a member of the Neuropeptide Y family, which is made up of 36 amino acid residues [18]. It is a highly conserved peptide among vertebrates, such as fish, and it is known to be among the most effective orexigenic agents in mammals [18,19].

As one of the most studied appetite-regulating peptides in fish, neuropeptide Y has been known to be an intricate peptide that regulates numerous physiological processes in several vertebrates, including food intake, circadian rhythms, and reproduction [14,15,20,21]. Since its first extraction from a mammalian brain [22], countless studies have demonstrated that Npy is synthesized in the central nervous system and other peripheral tissues of mammals and several teleost fish species, with the highest expression in the brain [13,23,24]. On several accounts, Npy has been proven to increase food intake in fish species, including goldfish (*Carassius auratus*) [25], zebrafish (*Danio rerio*) [26], grass carp (*Ctenopharyngodon idellus*) [27], tiger puffer (*Takifugu rubripes*) [28], yellowtail (*Seriola quinqueradiata*) [29], and blunt snout bream (*Megalobrama amblycephala*) [30]. Additionally, Npy in fish species as in mammals is known to interact with some appetite-regulating genes, including cocaine and amphetamine-regulated transcript (CART) and ghrelin [31,32].

For further investigations on the regulation of the appetite-regulating peptide Npy in the spotted scat, the full-length cDNA was cloned. We examined the mRNA level in several tissues in the fish and performed a localization study of the *npy* gene in some selected regions in the brain of the spotted scat. We further investigated the effects on the expression levels of *npy* mRNA in the hypothalamus of the spotted scat over pre- and post-prandial, as well as long-term fasting and refeeding periods. Further, using in vitro and in vivo studies, we desired to determine the relation and influence that the *npy* gene has on the mRNA level of some targeted appetite-inducing and inhibiting genes that have been studied in fish. Our aim for conducting this research is to contribute to a fundamental study that will help develop the feed intake regulation in spotted scat and to improve growth and reproduction in fish, which will be of significance to the aquaculture industry.

## 2. Materials and Methods

### 2.1. Experimental Fish

Spotted scat was bought from the Dongfeng Market (Zhangjiang, Guangdong, China). The fish were anesthetized in a solution with 50 mg/L tricaine methanesulfonate (MS-222, Sigma, Saint Louis, MO, USA) before dissection. Different tissues, including the hypothalamus, pituitary, gonads, gill, heart, kidney, liver, spleen, stomach, intestine, and muscle, were collected, frozen in liquid nitrogen immediately, and stored at $-80\,^{\circ}\text{C}$ awaiting RNA extraction. Experimental fish for in situ hybridization, post-prandial, and long-term fasting and refeeding experiments were cultured at Donghai Island (Zhangjiang, China) from fertilized eggs.

The fish experiments were conducted in agreement with the relevant guiding principle and were permitted by the Animal Research and Ethics Committees of Fisheries College of Guangdong Ocean University, China.

### 2.2. Cloning and Sequence Analysis of the Npy Gene

Neuropeptide Y cDNA sequences from Atlantic salmon (*Salmo salar*) NM_001146681.1 and goldfish (*Carassius auratus*) NC_039286.1 were blasted to our transcriptome sequence data of the spotted scat. Primers were then designed based on the mRNA transcripts to flank the ORF and cloned as indicated by Mustapha et al. [33]. To test the expression of the *npy* gene, we designed primers on the cloned sequence. Multiple alignments of the amino acid sequences were performed using ClustalX (http://www.clustal.org/ (accessed on 15 January 2022)). The conserved regions of the putative amino acid sequences were predicted using SMART (http://smart.embl-heidelberg.de/ (accessed on 17 January 2022)). MEGA6 software was used to build a phylogenetic tree with the neighbor-joining method (NJ). The bootstrap method with 1000 replications was applied to test the phylogeny.

### 2.3. Npy Expression Patterns in Different Spotted Scat Tissues, RNA Extraction and cDNA Synthesis

To determine the mRNA level of *npy* in spotted scat, 11 tissues (hypothalamus, pituitary, gonad, gill, heart, kidney, liver, spleen, stomach, intestine, and muscle) were immediately removed and stored in liquid nitrogen.

Going by the manufacturer's protocol, RNA was extracted from each of the selected tissues from the spotted scat. RNA samples were then stored at −80 °C until analysis. Total RNA was isolated from the tissue samples using the Trizol reagent (Invitrogen, Carlsbad, CA, USA). First-strand cDNA synthesis was performed using the PrimeScript™ RT reagent Kit with gDNA Eraser (Takara, Beijing, China) with 1 μg total RNA. The quality and concentration of the RNA were assessed by NanoDrop 2000 (Thermo Scientific, Wilmington, DE, USA), and the integrity was assessed on 1% agarose gel. Total RNA was treated with RNase-free DNase I (Thermo Scientific Corp, Waltham, MA, USA). For reverse transcription, using the All-in-One First Strand cDNA Synthesis SuperMix for qPCR (One-step gDNA Removal) (TransGen Biotech, Beijing, China), a mixture of 20 μL, made up of 15 μL (1 μg RNA + RNase free water), 4 μL of all-in-one supermix for qPCR, and 1 μL of gDNA remover was added and incubated at 50 °C for 15 min and 85 °C for 5 min. The final concentration was diluted three times to give a total volume of 80 μL cDNA for use as templates in all PCR reactions.

### 2.4. Npy Gene Localization in the Brain of the Spotted Scat via In Situ Hybridization

To examine the distribution of *npy* in spotted scat brains, in situ hybridization was performed. Digoxigenin-labeled (DIG) RNA probes were synthesized by Servicebio Company (Wuhan, China). First, spotted scat brains were fixed with 4% paraformaldehyde at 4 °C overnight. After, the brains were dehydrated through gradient ethanol concentrations, and infiltrated with paraffin (Life Company, Shanghai, China). The brain was embedded in the Optimal Cutting Temperature (OCT) compound (Tokyo, Japan) and sectioned serially at 10 μm on a slicer RM2125RTS (Leica, Mannheim, Germany). The sections were thaw-mounted onto aminopropyl silane-treated glass slides and dried in an oven for less than 30 min. The RNA probe sequences, which covered the ORF sequence of spotted scat Npy cDNA, were amplified with the specified primers and spliced into a pGEM-T vector (Tiangen, Beijing, China). Finally, the in situ hybridization experiment was performed following the Enhanced Sensitive ISH Detection Kit II (AP) protocol (BOSTER, Wuhan, China).

### 2.5. Post-Prandial Research and Long-Term Feeding, Fasting, and Refeeding of Spotted Scat

Seven groups of the spotted scat (*n* = 10/group), both males and females with average body weight and length 170 g and 17 cm, respectively, were used to perform the post-prandial research. The fish were allowed to acclimatize to the feeding time and the fiber tank for about 10 days. Samples were taken three hours (h) before feeding (−3 h), one hour before feeding (−1 h), the specific feeding time (0 h), one hour after feeding or being unfed (+1 h), and three hours post-feeding or being unfed (+3 h). A total of 6 out of the 10 fish in each group were selected randomly, and the hypothalamus of each was taken at the specific sampling periods and frozen immediately at −80 °C until RNA extraction.

For the long-term fasting and refeeding experiment, spotted scat averagely weighing 190 g and 18.5 cm in length were divided into two groups (*n* = 30; feeding and 35; fasting and refeeding). Similarly, they were left to acclimatize to a specific feeding time (10 a.m.) and the environment. Both the fasting and feeding groups were sampled on specific days (0–7), while the other fasted group was fed on days 8 and 10. Six fish from the unfed and refed group were randomly selected, while four fish were selected from the fed group, and the hypothalamus of each fish was collected on specific days, and stored at −80 °C until RNA extraction.

### 2.6. Npy Peptide Synthesis and In Vitro Incubation of the Spotted Scat

Spotted scat matured Npy peptide, as indicated in Figure 1A, was commercially synthesized in aminated form by GL Biochem limited (Shanghai, China). The Npy peptide was up to 95% pure according to high-performance liquid chromatography (HPLC). The peptide was dissolved in fish saline to 1, 10, and 100 ng/mL and stored at −20 °C.

Twelve fish, averagely weighing 140 g and 14.1 cm in length, were anesthetized with MS-222, and their hypothalamus was dissected and washed three times with M199 medium. The tissue fragments were transferred onto a 24-well culture plate with M199 medium amended with penicillin and bovine serum. After 2 h of pre-incubation at 25–28 °C with 5% $CO_2$, the medium was removed, and the fragments were washed twice with serum-free M199. The medium was removed and replaced with a fresh medium containing graded concentrations of Npy peptide (1, 10, and 100 ng/mL). The control group was tested at the same time without the Npy peptide (0 ng/mL). After 6 h of incubation, the tissue fragments were collected for RNA extraction as described above. Quantitative real-time PCR was used to measure their relative expressions with specific primers, as described in the "Real-time PCR analysis" section.

### 2.7. Npy Peptide In Vivo Injection of the Spotted Scat

A total of 24 fish (grouped into 4), with average body weight (BW) and length of 145.8 g and 13.36 cm, respectively, were intraperitoneally (i.p.) injected with saline-diluted concentrations of the spotted scat Npy peptide (1, 10, and 100 ng/g BW). Normal saline was intraperitoneally injected at the same time, serving as the control group (0 ng/g BW). The hypothalamus and pituitary were collected after 6 h of injection for RNA extraction, as described above. qPCR evaluation was used to evaluate the expression of other appetite-regulating genes in the hypothalamus, all with specific primers.

### 2.8. Quantitative Real-Time PCR (qPCR)and Statistical Analysis

We determined the mRNA level of *npy* in selected spotted scat tissues, pre- and post-prandial variations, the effects of long-term fasting and refeeding on the expression of the *npy* gene, and the in vitro and in vivo analyses of Npy in spotted scat hypothalamus using quantitative PCR (qPCR). RT-qPCR was performed in triplicate for each sample on a CFX96 Real-Time PCR Detection System (Bio-Rad, Indianapolis, IN, USA) in 20 μL reactions, containing the following components: 2 μL cDNA (template), 10 μL SYBR Premix EXTaq (TaKaRa, Shiga, Japan), 0.8 μL of each primer (forward and reverse), and 6.4 μL RNase free water. The PCR parameters were 40 cycles at 95 °C for 5 s, 60 °C for 30 s, followed by a dissociation curve analysis of 5 s per step from 65 to 95 °C. All samples for relative mRNA level analysis were run once in a single assay. The $2^{-\triangle\triangle CT}$ method was used to analyze the relative gene expression.

All PCR primers that were used in this research are listed in Table 1.

When necessary, data derived from this study were expressed as the mean ± SEM (standard error of the mean). Significant differences in the data among groups were determined at some point by the student's *t*-test and/or one-way analysis of variance (ANOVA), followed by Duncan's post hoc test. A likelihood level below 0.05 ($p < 0.05$) was used to specify significance. All statistical analyses were achieved using Statistical Package for the Social Sciences (SPSS) 19.0 (SPSS, Chicago, IL, USA).

**Table 1.** Genes and primer sequences used for cloning, tissue distribution, and in situ hybridization experiment.

| Gene and Primer Orientation | Primers Sequence (5′–3′) | Application |
|---|---|---|
| *Npy*-F | GCAAACCTCAGCCTAACCCTCT | ORF/ cloning |
| *Npy*-R | ACTGTAAGCAGTTGTGACTGTAGC | |
| *Npy*-TD-F1 | ATACCCGGTGAAACCCGAGAAC | Tissue Distribution (Gene |
| *Npy*-TD-R1 | GCGTGTCTGTGCTTTCCTTCAA | expression) |
| *Npy*-ish-F1 | taatacgactcactatagggATACCCGGTGAAACCCGAGAAC | |
| | | In situ hybridization |
| *Npy*-ish-R1 | taatacgactcactatagggGCGTGTCTGTGCTTTCCTTCAA | |
| *β -Actin*-F | GAGAGGTTCCGTTGCCCAGAG | House-keeping gene |
| *β -Actin*-R | ACGCCAACACTGTGCTGTCTG | |
| *Apelin*-F | ATGAATGTGAAGATCCTGACG | |
| *Apelin*-R | CTAGAACGGCATGGGTCCC | |
| *Ghrelin*-F | CTCAGCCCTTCACAGAAACCTC | *Npy* in vitro and in vivo |
| *Ghrelin*-R | ATCTCCTGCAACGCCACATCGT | experiment analysis on other |
| *Pyy*-F | GGAACACTGGCAGATGCCTACC | appetite-regulating genes |
| *Pyy*-R | TCTGTTGCTGTCGCCACCAA | |
| *Cart*-F | CAGCGCCGCTGGAGCCCAAGTGT | |
| *Cart*-R | TTCTTTTCCCACAGAGGCAGTC | |
| *Cck*-F | TGCTCGAAACAGCCTGAACCAG | |
| *Cck*-R | AAAGTCCATCCAGCCGAGGTAA | |

Note: taatacgactcactataggg (T7) is a promoter for in situ hybridization.

## 3. Results

*3.1. Cloning and Sequence Analysis of the Npy Gene*

The Npy peptide was cloned for the first time in the spotted scat. It contained a 300-bp ORF that encodes a 99-amino acid (aa) protein (prepro-Npy), which was composed of a signal peptide of 28 aa, a mature 36 aa peptide, 3 aa (GKR) forming the amidation-proteolytic site, and a 31 aa-spacer region of the NPY precursor (known as CPON, carboxy-terminal peptide of neuropeptide Y in Cerdá-Reverter et al. [17]) (see Figure 1A). A comparison of the spotted scat Npy amino acid sequence was analyzed with that of other species (Figure 1B).

*3.2. Phylogenetic Analysis of Npy*

The Npy phylogenetic tree was built using the neighbor-joining method. The amino acid sequences of spotted scat Npy were aligned with that of other organisms. These organisms included some teleost fish species, non-teleost, human, rodent, and chicken (see Figure 2A). Our results show that the spotted scat Npy is over 60% highly conserved with other vertebrates. The percentage similarities of the spotted scat Npy with other organisms were given in Figure 2B; where the Npy of spotted scat had the highest and lowest percentage similarities with *Pagrus major* and *Clarias gariepinus*, respectively.

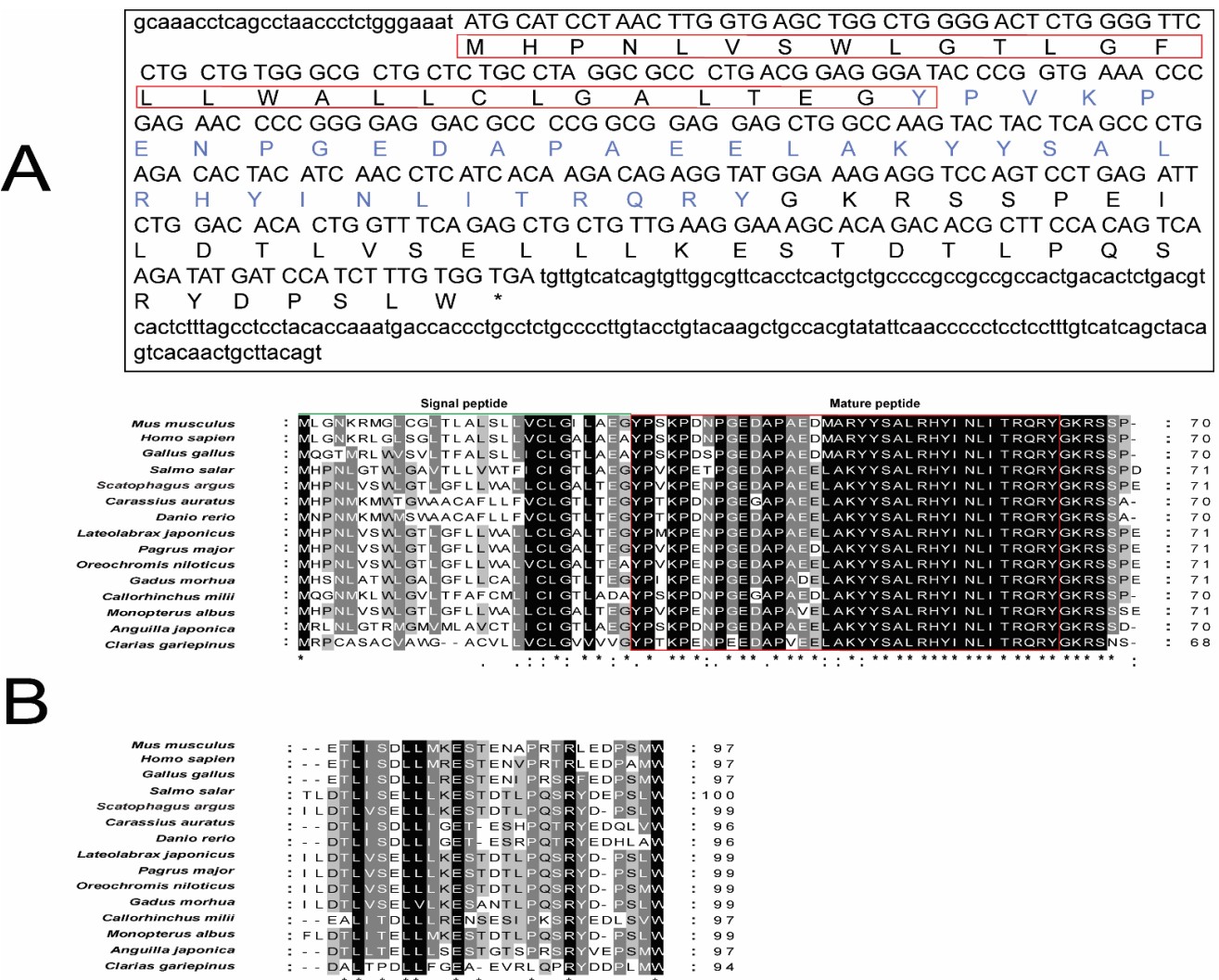

**Figure 1.** (**A**) Nucleotide and deduced amino acid sequence of spotted scat *Npy*. The supposed signal peptide of Npy is boxed, and the matured peptide is written in blue. The stop codon is indicated by an asterisk (*). (**B**) Comparison of the amino acid sequence of *Scatophagus argus* Npy against that of *Mus musculus* (NP_075945.1), *Homo sapien* (NP_000896.1), *Gallus gallus* (NP_990804.1), *Salmo salar* (NP_001140153.1), *Carassius auratus* (XP_026087265.1), *Danio rerio* (NP_571149.1), *Lateolabrax japonicus* (AIK01745.1), *Pagrus major* (BBA20652.1), *Oreochromis niloticus* (XP_003448902.1), *Gadus morhua* (XP_030203389.1), *Callorhinchus milii* (ACF22970.1), *Monopterus albus* (AEX97166.1), *Anguilla japonica* (AFN84517.1), and *Clarias gariepinus* (AOZ35531.1). (*) indicates positions that have a fully conserved residue, (:) indicates conservation between groups of strongly similar properties, and (.) indicates conservation between groups of weakly similar properties. Green line indicates the signal peptide of each organism whiles the boxed area in red indicates the matured peptide of the spotted scat Npy and the Npy of other organisms.

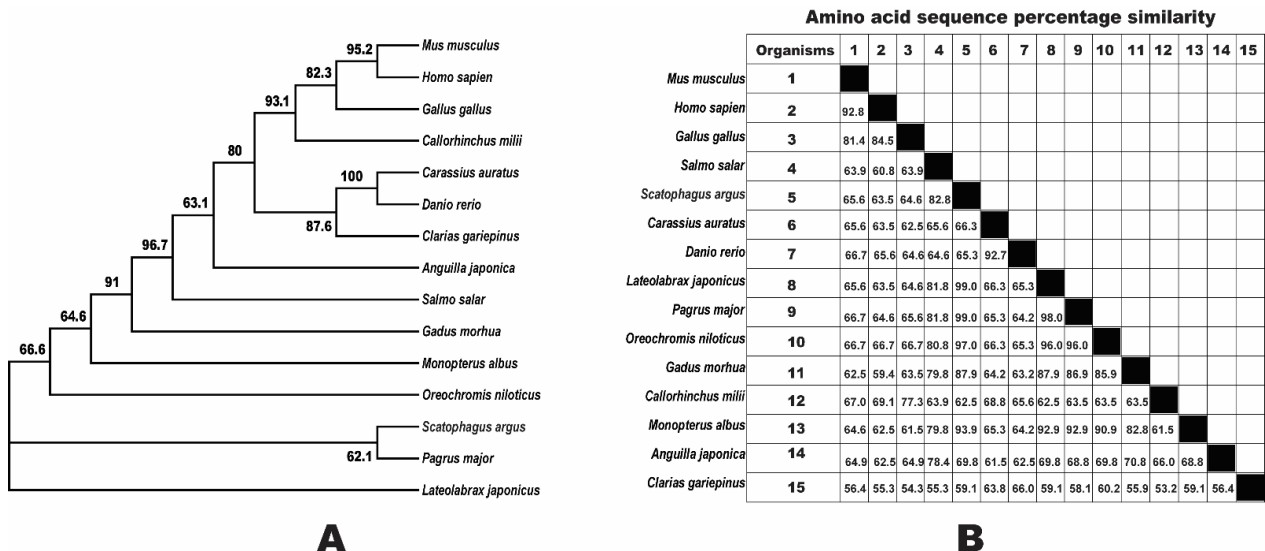

**Figure 2.** (**A**) Phylogenetic tree spotted scat Npy peptide with Npy of *Mus musculus* (NP_075945.1), *Homo sapien* (NP_000896.1), *Gallus gallus* (NP_990804.1), *Salmo salar* (NP_001140153.1), *Carassius auratus* (XP_026087265.1), *Danio rerio* (NP_571149.1), *Callorhinchus milii* (ACF22970.1), *Monopterus albus* (AEX97166.1), *Anguilla japonica* (AFN84517.1), *Clarias gariepinus* (AOZ35531.1), *Lateolabrax japonicus* (AIK01745.1), *Oreochromis niloticus* (XP_003448902.1), *Gadus morhua* (XP_030203389.1), and *Pagrus major* (BBA20652.1). (**B**) Amino acid sequence percentage similarity of Npy peptide in spotted scat with that of other organisms.

### 3.3. Npy Gene in Spotted Scat Tissues

Our research determined the distribution characteristics of *npy* in selected spotted scat tissues using RT-qPCR. From the highest to the lowest, *npy* mRNA was identified in the pituitary, spleen, muscle, intestine, testis, gill, liver, hypothalamus, heart, kidney, and stomach in males, while in females, *npy* mRNA was detected in the pituitary, spleen, muscle, intestine, heart, hypothalamus, liver, gill, stomach, kidney, and ovary (Figure 3).

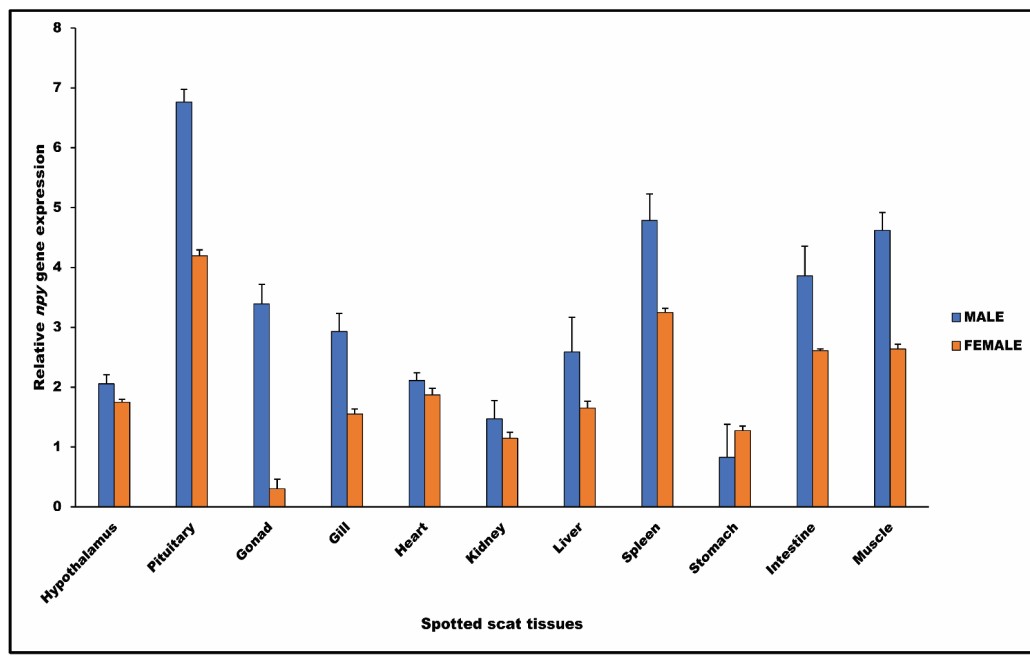

**Figure 3.** Tissue expression arrays of *npy* mRNA in the spotted scat. Amplification of β-actin was used as the in-house regulator. Data are given as mean ± standard error of the mean.

### 3.4. Localization of Npy mRNA in Spotted Scat Brain

The in situ hybridization experiment was carried out in the brain of the spotted scat (Figure 4i) for the localization of the *npy* gene. Two regions of the spotted scat brain; the telencephalon and optic tectum, were selected for this experiment, as shown in Figure 4i. In the antisense probes, mRNA-expressing cells filled with *npy* signals were detected in these regions of the brain; the telencephalon (Tel) (*) and optic tectum region (OT) (**), as shown in Figure 4(iiC, iiD). The sense probes of *npy* did not show any signal in respect to the brain regions. However, the representative region of the telencephalon was only provided as evidence of the negative control (Figure 4(iiA), with higher magnification in Figure 4(iiB)).

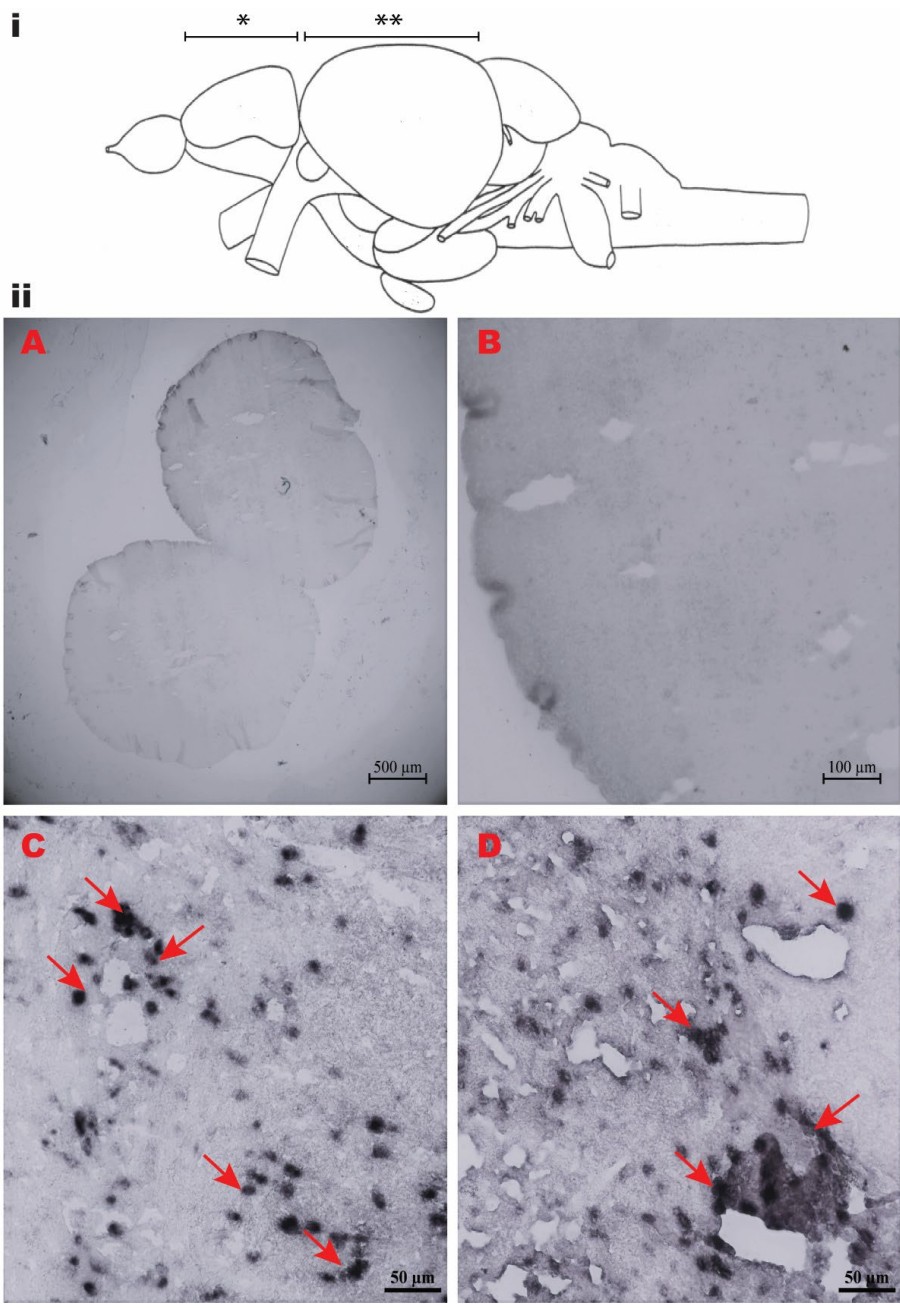

**Figure 4.** The localization of the *npy* mRNA-expressing cells in the brain of the spotted scat via in situ hybridization. Sense probe of the *npy* gene (**A**,**B**), antisense probe in the telencephalon (**C**), antisense probe in optic tectum region (**D**). Scale bar indicates (**A**: 500 μm) (**B**: 100 μm) and (**C**,**D**: 50 μm). Note: No signal was detected in the sense probe against the antisense probe. The arrows in (**C**,**D**) indicate positive signals of the *npy* gene.

### 3.5. Pre and Post-Prandial and Long-Term Effects of Feeding, Fasting, and Refeeding on Npy mRNA Level in the Spotted Scat

The result from our pre and post-prandial experiment demonstrates the expression pattern of *npy* mRNA in the hypothalamus of spotted scat within a short period of time (see Figure 5). There was a significant increase in the expression level of *npy* mRNA three hours after being starved (+3 h), as compared to one and three hours before feeding time (−1 h and −3 h), the specific feeding time (0 h), and one hour into the starvation period (+1 h). On the other hand, there was no significant difference in the expression of *npy* mRNA one and three hours after being fed (+1 h and +3 h). However, we identified that the expression of *npy* mRNA in the spotted scat hypothalamus one hour (+1 h) after being fed was significantly higher compared to one hour of being starved (+1 h), whereas there was no significant difference in the *npy* mRNA level in the hypothalamus of spotted scat +3 h after being fed or being starved, even though the expression in the unfed group was higher in that of the fed group.

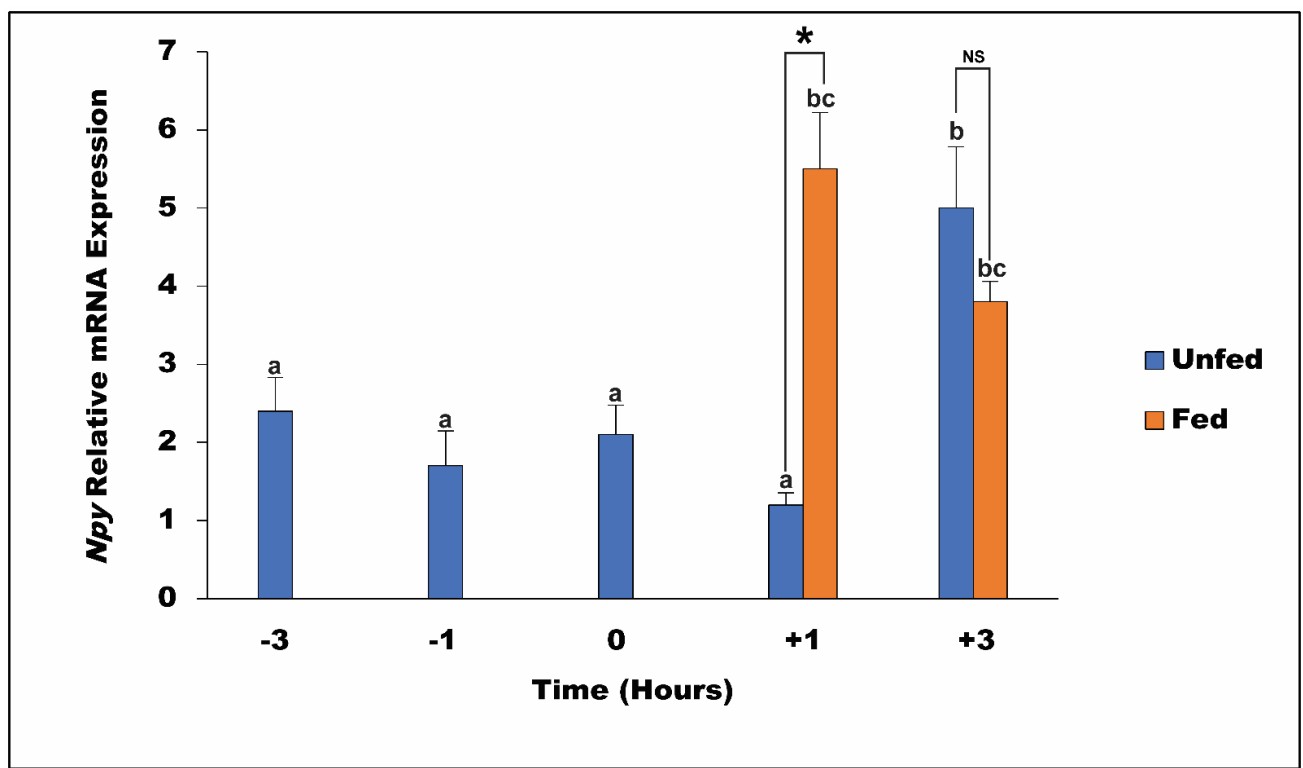

**Figure 5.** Pre- and Post-prandial effects on *npy* mRNA levels in the hypothalamus of *Scatophagus argus*. Bars with different letters represent significant differences between experimental groups, while the asterisk represents the significant difference between groups at a specific time (* $p < 0.05$). NS means non-significant.

To determine whether the denial of food for a relatively long time (days) influenced the expression of the *npy* gene in the hypothalamus of *Scatophagus argus*, we compared *npy* mRNA expressions in the hypothalamus between fed, unfed, and refed groups. Our research indicated that there was a significant difference in the expression of *npy* in spotted scat hypothalamus during the fasting period. On the second and fifth days of this study, there was a significant difference between the expression of *npy* mRNA in the fed (decrease) and unfed group (increase). There was no significant difference in the expression of *npy* mRNA between fed and unfed groups on the seventh day. However, after refeeding on the 8th and 10th day, the was a significant increase (8th day) of *npy* expression in the hypothalamus of the spotted scat compared to on the 10th day, see Figure 6.

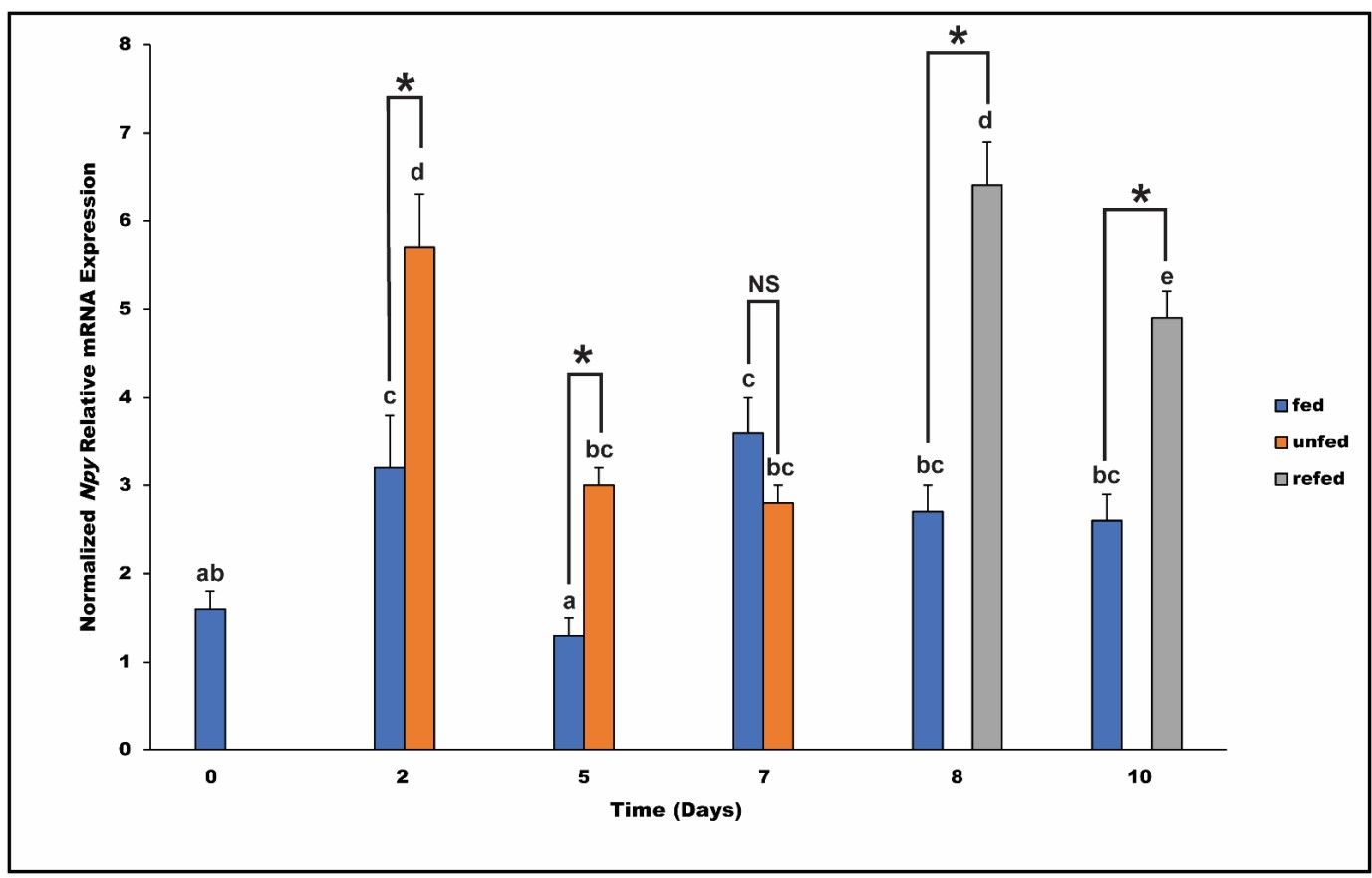

**Figure 6.** Fasting and refeeding effects on the level of *npy* mRNA in the hypothalamus of *Scatophagus argus*. Bars with different letters represent significant differences between experimental groups. Asterisks represent significant differences between the groups at the same time point; * $p < 0.05$. NS means non-significant.

### 3.6. In Vitro and In Vivo Effects of Npy on Hypothalamus Expression of Other Appetite-Regulating Genes in Spotted Scat

The hypothalamic expressions of selected appetite-regulating genes were concurrently measured by RT-qPCR technology in spotted scat six hours after Npy treatments (incubation and injections) (Figure 7). Our results demonstrate that the highest concentration of Npy peptide used for this study (100 ng/mL) could significantly promote the expression of *apelin* mRNA in the spotted scat in vitro while both moderate and highest Npy peptide concentrations (10 and 100 ng/gBW) could influence an increase in the mRNA level of *apelin* via the in vivo experiment (Figure 7A). In Figure 7B, we discovered that spotted scat hypothalamus incubated with an Npy concentration of 100 ng/mL could significantly promote the expression of *ghrelin* after 6 h while none of the concentrations of Npy peptide used for injection (1, 10, and 100 ng/gBW) could influence the expression of *ghrelin* in the spotted scat hypothalamus after 6 h. The results shown in Figure 7C revealed that all concentrations of the Npy peptide used for the incubation of the hypothalamus (1, 10, and 100 ng/mL) could significantly inhibit the expression of *pyy* mRNA in spotted scat, but only one concentration of Npy peptide used for injecting the fish (100 ng/gBW) could attenuate the hypothalamic expression of *pyy* in the fish. In Figure 7D, E, all Npy peptide concentrations used for the incubation of the hypothalamus could significantly suppress the expression of *cart* and *cck* in the spotted scat six hours after treatments, while none of the Npy peptide concentrations (1, 10, and 100 ng/gBW) could significantly influence the mRNA expressions of *cart* and *cck* in the hypothalamus of spotted scat via in vivo experimental studies.

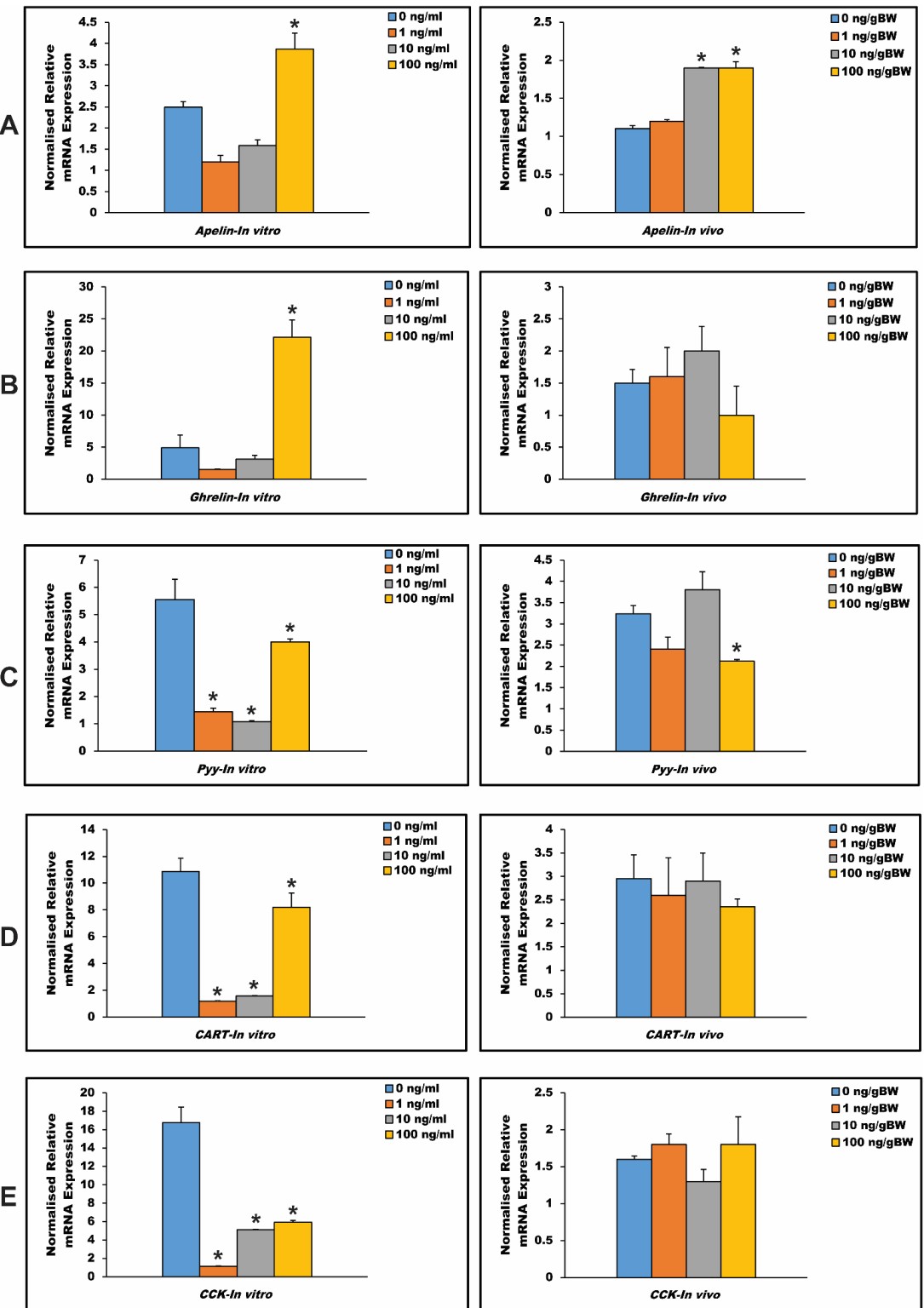

**Figure 7.** In vitro and in vivo effects of Npy treatment with peptide concentrations of 1 ng/mL–1 ng/gBW, 10 ng/mL–10 ng/gBW, and 100 ng/mL–100 ng/gBW on the quantitative expression of other appetite-regulating peptides in the hypothalamus of spotted scat. (**A**) Apelin, (**B**) Ghrelin, (**C**) Pyy (**D**) CART, and (**E**) CCK. Data are given as mean ± SEM. Bars with asterisks specify groups that significantly differ. The significant difference was identified using one-way ANOVA (Post Hoc Tests, LSD, followed by Duncan tests, $p < 0.05$). Significant difference was provided in reference to the control/saline group (0 ng/gBW–0 ng/gBW).

## 4. Discussion

This report describes the characterization, tissue distribution, gene localization, and the expression effect of *npy* over post-prandial and fasting and refeeding. The *npy* gene is an imperative appetite-stimulating factor. It has been proven to increase feed intake in fish (for example; [32]).

### 4.1. Cloning and Sequence Analysis of the Npy Gene

In the present study, the cDNA sequence of Npy was cloned from the spotted scat (*Scatophagus argus*) for the first time. There is a higher level of conservation between spotted scat Npy and other teleost fish species (vertebrates) amino acid sequences, which designates that Npy is among the highly conserved neuropeptides in vertebrate evolution [34], signifying the importance of its physiological function [27]. To disclose the molecular phylogenetic position of Npy, a phylogenetic tree was created by the neighbor-joining method, as shown in Figure 3. Our results from this prove that spotted scat Npy is highly conserved (>60%) when compared with other vertebrates. These results are in line with many studies on other fish species, including the Northern Snakehead (*Channa argus*) [35], blunt snout bream (*Megalobrama amblycephala*) [30], and the *Schizothorax davidi* [36].

### 4.2. Npy Gene Expression in Spotted Scat Tissues

The expression of *npy* mRNA in spotted scat tissues, including the brain (hypothalamus), pituitary, gonads, gill, heart, kidney, liver, spleen, stomach, intestine, and muscle, confirms the studies by several researchers that *npy* is mainly expressed in the brain and some peripheral tissues of numerous fish species [13,30,37–39].

In some fish species, such as goldfish, Atlantic cod, and catfish, *npy* expression in the forebrain is influenced by food availability and displays peri-prandial changes, indicating that forebrain *npy* is involved in the regulation of food intake [13,25,40]. Relatively high expression levels of *npy* in the brain (hypothalamus) of spotted scat suggest that *npy* might also be involved in regulating food intake in this species. We found that *npy* mRNA is expressed in the gut (specifically, the intestine), heart, kidney, and gonads. *Npy* mRNA is expressed in the gastrointestinal tract of several fish species, including lamprey, Atlantic cod, Brazilian flounder (*Paralichthys orbignyanus*), Chinese perch (*Siniperca chuatsi*), blunt snout bream (*Megalobrama amblycephala*), and yellowtail fish (*Seriola lalandi*) [13,29,30,37,39,41]. The presence of *npy* mRNA in the intestine suggests that, as in mammals and other fish species, spotted scat Npy acts as a brain–gut peptide and may influence gastrointestinal processes, as indicated by Deng and colleagues [36].

Our observations that *npy* mRNA is present in spotted scat gonads are in line with previous studies showing that *npy* mRNA is expressed in fish gonads [13,36,42] and that *npy* affects sexual behavior in goldfish [43] and aids in the release of gonadotropin-releasing hormone in catfish [44]. With that, we suggest that the presence of *npy* mRNA in the gonads might also play a role in the control of reproductive events in spotted scat. Spotted scat *npy* mRNA is also expressed in the gill, kidney, and heart. Studies confirmed *npy* mRNA level in the kidney of pufferfish [45] but not in the kidney or heart of catfish [42], but it was expressed in the kidney of Atlantic cod (*Gadus morhua*) [13]. It was also expressed in the gill of the snakehead fish (*Channa argus*) [35], suggesting that spotted scat *npy* could be involved in the mechanical process aiding in respiration. *Npy* increases heart rate in dogfish [46], suggesting that *npy* might be involved in the cardiac function of fish. Additionally, the *npy* mRNA level in the kidney of fish suggests that *npy* might be involved in fish osmoregulation [13]. Furthermore, *npy* mRNA was identified in the muscle of the spotted scat. In 2007, Liang and colleagues also identified *npy* mRNA in the Chinese perch muscle. In their article, they clarified that drawing functional conclusions based on the anatomical study in a single species could be redundant, irrespective of how consistent *npy* mRNA expression in this species is to others [39].

*4.3. The Npy Gene in Spotted Scat Brain*

Several studies using fish species as research models have highlighted the expression of *npy* mRNA in various sections of the brain, including the telencephalon and optic tectum, using either qPCR or RT-PCR; e.g., Refs. [13,29,36,47]. As evidence for our research, we selected two regions on which we performed a gene localization experiment via in situ hybridization.

The in situ hybridization experiment has revealed the mRNA level of the *npy* gene in various regions of the forebrain and midbrain regions of the spotted scat, including the telencephalon and the optic tectum. Our result is in line with research on goldfish, where in situ hybridization and other experimental studies demonstrated that *npy* mRNA was expressed in the forebrain and midbrain regions of goldfish. These include the telencephalon, olfactory bulb, the thalamic regions, and the optic tectum and locus coeruleus, respectively [40,43]. It is also consistent with results from the sea bass (*Dicentrarchus labrax*), where positive signals of *npy* were determined in the telencephalon (subsections) and other parts of the brain [48]. Additionally, *npy*-immunoreactive cells were determined in several sections of the catfish (*Clarias batrachus*) (see Gaikwad et al. [49] and references therein). Positive hybridization signals of *npy* located in the telencephalon and the optic tectum of the spotted scat suggest the involvement of the brain in various physiological processes, such as food intake, circadian rhythms, and reproduction [20,21]. As described by Sudhakumari and colleagues, the *npy* mRNA level in the optic tectum, in addition to the cerebellum and thalamus (OCT), has a role to play in the feeding behavior of fishes [20]. Thus, positive signals of *npy* in the optic tectum of the spotted scat might have a similar function in the fish's behavior towards feeding. The positive signals of *npy* in these regions of the spotted scat brain suggest a wide distribution of the gene across the brains of fish species [28,50].

*4.4. Post-Prandial and Fasting and Refeeding Influence on Npy Expression in Spotted Scat Hypothalamus*

Studies in the past have indeed proven that Npy plays a key role in the regulation of food intake in many organisms, including teleost fish species [23]. Under normal circumstances, appetite-regulating peptides with orexigenic roles, including Npy, Ghrelin, and Apelin, have pre-prandial increase and post-prandial decrease in expression, e.g., Refs. [15,30,51], while those with anorexigenic roles in feed intake regulation, such as Pyy, CART, and CCK, have post-prandial increase and pre-prandial decrease in expression, e.g., Refs. [52–54].

In an hourly-based experiment, data obtained from this research reveals that one hour after the feeding time, the expression of *npy* in the spotted scat hypothalamus was significantly lower in the unfed group than in the fed group ($p < 0.05$). On the other hand, there was no significant difference in the expression of *npy* in the hypothalamus of *scatophagus argus* three hours after the feeding time, even though the expression in the fed group was lower compared to the unfed group. This is contrary to what previous studies have indicated. For example, in research conducted on gibel carp (*Carassius auratus gibelio*), the hypothalamus expression of *npy* was significantly lower in the fed group than in the unfed group, one and three hours after feeding [47]. Further, in goldfish (*Carassius auratus*), *npy* expression levels in the brain one and three hours after feeding sharply decreased [25]. Similarly, the expression of *npy* in Ya-fish (*Schizothorax prenanti*) was significantly lower 0.5, 1.5, 3, and 9 h after feeding [15]. However, in Atlantic salmon (*Salmo salar*), data obtained showed that there was an influence of feeding on the *npy* expression level in the brain; a significant increase in the expression levels of *npy* 0.5 to 9 h after feeding [54], which is similar to our studies. All these indicate that the mechanism governing the expression of *Npy* differs with teleost fish species within a short time and may be clarified by the biological endocrine effects, as explained further by Boujard and Leatherland [55]. As such, further studies are much need.

With regards to the long-term fasting and refeeding, there was a highly significant increase of *npy* mRNA levels in the spotted scat hypothalamus before the 7th day of the fasting period (days 2 and 5). This is to clarify that it takes a few days (probably less than a week) of food deprivation to determine significant changes in the expression of the *npy* gene in spotted scat hypothalamus compared to other species, such as the channel catfish (*Ictalurus punctatus*); which takes three weeks [56]. This is in line with research conducted on *Schizothorax davidi*, where there was a significant increase in the expression of *npy* mRNA in the hypothalamus after five days of fasting, and the subsequent days after [36], unlike other fish species that had a significant increase in the expression of *npy* after being fasted for more than five days. These include the Northern snakehead (*Channa argus*) [35], zebrafish (*Danio rerio*) [26], yellowtail (*Seriola quinqueradiata*) [29], and the Brazilian flounder (*Paralichthys orbignyanus*) [37]. After refeeding, the results obtained showed that the *npy* mRNA level in the refed group was significantly higher than in the fed group ($p < 0.05$). This was in contrast with previous research studies. For instance, in fish species, such as the blunt snout bream (*Megalobrama amblycephala*) [30], the Northern snakehead (*Channa argus*) [35], *Schizothorax davidi* [36], and the gibel carp (*Carassius auratus gibelio*) [47], the expression level of *npy* mRNA in the refed groups was significantly lower than in the fed group. This could be as a result of fish specificity, as well as the anticipation to eat more food than expected.

### 4.5. The Appetite Regulatory Role of Npy via In Vitro and In Vivo Experimental Studies

The regulatory role of Npy on key appetite-regulating peptides (Apelin, Ghrelin, Pyy, CART, and CCK) was investigated in the spotted scat through in vitro and in vivo experiments. There is a wide mRNA level of these genes in the central nervous system, as well as other peripheral tissues in many fish species. This indicates the structural linkage between the systems in which these peptides are located. The hypothalamus is known to be a key central regulator of feed intake and energy balance in fish species. It is known to integrate the brain and peripheral signals associated with the consumption of food and digestion, metabolism, and energy storage [51,57].

After examining the hypothalamic interaction of the *npy* gene with other appetite regulators in the spotted scat hypothalamus, our studies via in vitro and in vivo experiments established that there is a local interaction that exists between Npy and other appetite-regulating peptides in the spotted scat. This has already been indicated by Volkoff and colleagues, stating that Npy closely interacts with other appetite-regulators in fish [51]. Npy significantly upregulated the mRNA level of *apelin* in the hypothalamus of spotted scat in vitro and in vivo. It significantly upregulated the mRNA level of *ghrelin* in vitro but failed to significantly influence *ghrelin* in vivo. In vitro and in vivo analyses proved that Npy significantly decreased the expression of *pyy* in spotted scat brains. On the other hand, the results revealed that Npy significantly downregulated the expression of *cart* and *cck* via in vitro. However, it failed to significantly downregulate the mRNA level of *cart* and *cck* in vivo. The difference in results may be related to species-specific reasons, peptide concentration, time-based reasons, gene isoform, conditions, and/or treatment methods.

To date, there is limited information in the literature on the modulation of Npy in the hypothalamic expression of the selected appetite-regulating peptides in fish species. There are few reports available where appetite-inducing peptides, such as Ghrelin and Apelin, have been known to interact; downregulate, upregulate and/or have interacting orexigenic effects with other appetite-regulating peptides in the brain (whole and/or in parts) and peripheral tissue of some fish species, including goldfish [58,59], common carp (*Cyprinus carpio*) [60], and cavefish (*Astyanax fasciatus mexicanus*) [61]. Reports reveal that Npy, similar to Apelin and Ghrelin, promotes the intake of food in fish [51,58,62–65]. These peptides work hand-in-hand to persuade the orexigenic expression of each other in order to stimulate the consumption of food in fish species. Research revealed that Apelin treated hypothalamus significantly caused an increase in the expression of *npy* in the common carp [60]. Further, in goldfish, Npy stimulated the increase of *apelin* mRNA

level. From this same research, Apelin was demonstrated to stimulate *npy* expression in the hypothalamus [66]. Confirming their dependency on expression to induce food intake in fish. With less information on the interaction, Npy has on the hypothalamic mRNA level of *ghrelin* in fish; a review has revealed that ghrelin triggers the expression of AgRP and/or Npy neurons, which then aids in stimulating food intake [67]. All things considered, we speculate that the expression of the *npy* gene has the tendency to stimulate the consumption of feed and feeding behavior by upregulating the hypothalamic orexigenic roles of *apelin* and *ghrelin* in spotted scat.

The presence of the *npy* gene, on the other hand, significantly downregulates *cart*, *cck*, and *pyy* in the hypothalamus of spotted scat. This insinuates that Npy peptide effects designate that this peptide wields a modulation of the expression of hypothalamic anorexigenic genes in the spotted scat. Thus, the administration of Npy in the spotted scat can inhibit the anorexigenic roles of these appetite-inhibiting peptides in feed intake and vice versa. This can be confirmed from similar research, where ghrelin (an orexigenic factor) suppressed the anorexigenic roles of some appetite-inhibitors, including CCK and CART in goldfish [58]. In conformity, the administration of CART [68] and CCK [69] has been identified to inhibit the orexigenic effects of the *npy* gene on food intake in goldfish. While there has not been any study with that on the Pyy peptide in fish, in vitro analyses revealed a decrease in the hypothalamic expression of the *npy* gene when a rat was treated with Pyy [70]. Accurately, the fact that Npy attenuates important anorexigenic genes is in agreement with its nature as a potent orexigenic gene in several fish species [14,28,31,32,35,51,71].

The interaction of Npy and other appetite-regulating peptides was assessed six hours after the peptide incubation and injection. Further studies should be performed to determine the effects of Npy on orexigenic and anorexigenic genes under shorter (1–3 h) and longer (7–12 h) incubation and injection periods.

*4.6. Npy as Growth Control Peptide in Spotted Scat*

The intake of food and growth are inseparably linked. The hypothalamus–pituitary–liver axis regulates growth in animals, including fish species. The hypothalamus secretes growth hormone-releasing hormone (GHRH); it then acts on the pituitary to influence the secretion of growth hormones (GH) [60]. The availability of food, in terms of quality and quantity, and the optimization of food intake play important roles in improving growth, body composition, and the reproduction of fish, which are key objectives in intensive fish culture [72]. The significant increase in the expression of *npy* in the spotted scat hypothalamus during the fasted period and, most importantly, after the refeeding period proves that Npy is a key candidate in the regulation of growth in the spotted scat. Feed, one of the most commanding gestures outside a fish's body, arouses the feeding behavior and growth in fish [73,74]. The ability of a fish to continually feed indicates the activeness or upregulation of key orexigenic factors in the system of the fish. These orexigenic factors, including Npy, aid in the stimulation of growth hormone in fish. For instance, in goldfish, it was revealed that Npy stimulates the expression of growth hormone [75]. That is to say, the higher expression of *npy* regulates the expression of growth hormone in fish, thereby aiding in the growth of fish [23].

**5. Conclusions**

The purpose of this study was to identify the Npy peptide in the spotted scat for the first time. We revealed that Npy in spotted scat is highly conserved, as in other vertebrates, most especially with other teleost fish species. Aside from peripheral tissues, the spotted scat *npy* gene was highly expressed in the brain, depicting that it plays a significant role in the regulation of food intake. Additionally, the expression of *npy* in several tissues of the spotted scat suggests that the gene could be involved in a variety of physiological purposes in various organs. After exploring and observing the expression profiles of *npy* during feeding and fasting, we conclude that Npy may be an appetite-inducer in this species. Moreover, our results from the in vitro and in vivo studies conform to previous studies,

indicating that the Npy peptide locally interacts with other appetite-regulating peptides in the spotted scat. This fundamental study generally provides a foundation for future research on the regulating mechanisms and functions that trigger the role of Npy on food intake in teleost fish.

**Author Contributions:** Conceptualization, D.A., H.C. and G.-L.L.; methodology, D.A., Y.W. and H.C.; investigation, D.A., U.F.M., C.B.N., Y.W. and Z.L.; writing—original draft preparation, D.A.; writing—review and editing, D.A., U.F.M. and H.C.; funding acquisition, H.C. and G.-L.L. All authors have read and agreed to the published version of the manuscript.

**Funding:** This article was supported by grants from the Guangdong Basic and Applied Basic Research Foundation (2019A1515010958), the Key Research and Development Program of Guangdong (2021B202020002), the talent team tender grant of Zhanjiang marine equipment and biology (2021E05035), the National Natural Science Foundation of China (41706174), and the Southern Marine Science and Engineering Guangdong Laboratory (Zhanjiang) (ZJW-2019-06).

**Institutional Review Board Statement:** The study was conducted in accordance with the guidelines of the Animal Research and Ethics Committee of the Institute of Aquatic Economic Animals, Guangdong Ocean University, China (201903004). The study did not require approval by the local ethics committee.

**Conflicts of Interest:** The authors declare no conflict of interest.

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
