# Peer review of "Neuropeptide Y in Spotted Scat (Scatophagus Argus), Characterization and Functional Analysis towards Feed Intake Regulation"

_fishes, doi:10.3390/fishes7030111_

Round 1

Reviewer 1 Report

The authors were the first to sequence Npy in spotted scat and investigate the tissue distribution of Npy mRNA expression. They also discussed the role of Npy in food intake based on data on changes in gene expression in fast/fed fish. This paper provides new insights into Npy in spotted scat and is worthy of publication in Fishes with some modifications.

  1. figure 3: how many samples were used for RT-PCR analysis of Npy expression in spotted scat tissue? How many times were the experiments repeated? In order to compare the expression levels of Npy genes in tissues, quantitative analysis with PCR bands should be required with graphs.

  1. in situ hybridization: the author shows the signal of the antisense probe in the telencephalon and midbrain. The signal is very clear, but it is not clear where in the telencephalon and midbrain the signal is detected. A schematic of a transverse section of the midbrain and telencephalon would be helpful to better understand the origin of the photograph. Figure 4 is important to show the brain distribution of Npy in the spotted scat.

  1. Figures 3 & 4: Were the fish used fasted or fed? If fasted, for how long?

  1. figure 7 shows the interaction between Npy and appetite-regulated genes in vitro and figure 8 shows the interaction between Npy and appetite-regulated genes in vivo. On the other hand, the expression pattern of appetite-regulated genes is different between Figure 7 and Figure 8. The authors explain that this difference is due to peptide concentration, gene isoforms, etc. In my opinion, since the in vitro results do not reflect the in vivo experiments, the authors should remove Figure 7 to better show the interaction between Npy and other appetite-related genes. in vivo results alone are sufficient.

  1. Since AgRP and NPY are known to co-localize in neurons and regulate appetite, it is recommended to estimate AgRP mRNA levels if possible.

Author Response

Response to Reviewer 1

Q1: Figure 3: how many samples were used for RT-PCR analysis of Npy expression in spotted scat tissue? How many times were the experiments repeated? In order to compare the expression levels of Npy genes in tissues, quantitative analysis with PCR bands should be required with graphs.

Reply: 5 tissues each were used for RT-PCR and the experiment was repeated 3 times. we have made the quantitative analysis with PCR bands and added the graphs in the revised manuscript.

Q2: In situ hybridization: the author shows the signal of the antisense probe in the telencephalon and midbrain. The signal is very clear, but it is not clear where in the telencephalon and midbrain the signal is detected. A schematic of a transverse section of the midbrain and telencephalon would be helpful to better understand the origin of the photograph. Figure 4 is important to show the brain distribution of Npy in the spotted scat.

Reply: Thank you, the purpose of this particular experiment was to confirm the existence of the gene in the brain of the spotted scat. Much attention wasn’t given to specific sections in the telencephalon and optic tectum, subsequent studies will focus more on the specific regions that the signals were identified.

Q3: Figures 3 & 4: Were the fish used fasted or fed? If fasted, for how long?

Reply: Thank you for your suggestion. The fish were not fasted; they were fed on daily bases.

Q4: Figure 7 shows the interaction between Npy and appetite-regulated genes in vitro and figure 8 shows the interaction between Npy and appetite-regulated genes in vivo. On the other hand, the expression pattern of appetite-regulated genes is different between Figure 7 and Figure 8. The authors explain that this difference is due to peptide concentration, gene isoforms, etc. In my opinion, since the in vitro results do not reflect the in vivo experiments, the authors should remove Figure 7 to better show the interaction between Npy and other appetite-related genes. in vivo results alone are sufficient.

Reply: Thank you for your suggestion. We would like to maintain this results in the manuscript. This will help further research identify how some internal and/or external factors could influence the expression of genes in fishes.

Q5: Since AgRP and NPY are known to co-localize in neurons and regulate appetite, it is recommended to estimate AgRP mRNA levels if possible.

Reply: Thanks for your recommendation. The co-localization of AgRP and Npy in the regulation of appetite is well known. Hence, we opted to look for something new, thus, using the selected genes as targeted genes.

We appreciate for Reviewers’ works earnestly, and hope that the correction will meet with approval. Once again, thank you very much for your comments and suggestions.

Reviewer 2 Report

My comments are in the attached file.

Author Response

Response to Reviewer2

Q1: Line 17 : « The neuropeptide (Npy) gene is an intricate gene, regulating … ». Please replace this sentence by : « The neuropeptide (Npy) is an intricate neuropeptide, regulating … ». These functions are ensured by the peptide and not the gene. The authors are reminded that the rules for gene/protein nomenclature are as follows: Gene symbols are italicised, with all letters in lowercase (npy). Protein designations are the same as the gene symbol, but are not italicised; the first letter is in uppercase and the remaining letters are in lowercase (Npy). The entire manuscript needs to be revised accordingly.

Reply: Thank you for pointing this out, we have revised the manuscript accordingly.

Q2: Line 18 : « It’s ». please replace by : « It is ».

Reply: Thank you, it has been replaced.

Q3: Line 19 : « To enlighten the mechanism of the Npy gene ». Please replace by : « to enlighten the mechanism of action of Npy in … ».

Reply: Thank you, it has been replaced.

Q4: Line 46 : « … numerous appetite-regulating genes, including neuropeptide Y (Npy) … ». Please replace by : « … numerous appetite-regulating peptides, including neuropeptide Y (Npy) … ».

Reply: Thank you, it has been replaced.

Q5: Line 48 : Add the following reference to ref [16] : Neuropeptide Y family of peptides: structure, anatomical expression, function, and molecular evolution. Cerdá-Reverter JM, Larhammar D. Biochem Cell Biol. 2000;78(3):371-92.

Reply: Thank you, this reference has been added.

Q6: Lines 52-55 : « … appetite-regulating gene, neuropeptide Y … ». Please replace by : « … appetite-regulating peptide, Npy … ».

Reply: Thank you, it has been replaced.

Q7: Line 53 : Suppress « conjunctinonal »

Reply: Thank you, the word conjunctional has been deleted.

Q8: Line 56 : Please replace « expressed » by « produced » or « synthesized »

Reply: Thank you, ‘expressed’ has been replaced by ‘synthesized’.

Q9: Line 56, line 58, line 62 : Please replace « Npy » by « Npy »

Reply: Thank you, it has been replaced.

Q10: Line 65 : Replace « … gene Npy » by « … gene npy »: The word « gene » is suitable here, but the first letter is not in uppercase

Reply: Thank you, it has been replaced and well noted.

Q11: Line 71 : Replace « Npy gene » by « Npy »

Reply: Thank you, it has been replaced.

Q12: Line 107: Please replace « secluded » by « isolated ».

Reply: Thank you, it has been replaced.

Q13: Line 116: add a gap between « 80 » and « μl »

Reply: Thank you, it has been added as suggested.

Q14: Lines 152-57: The protocol is unclear. Please rephrase the paragraph.

Reply: Thank you, this paragraph has been rephrased as “Both the fasting and feeding groups were sampled on specific days (0-7) while the other fasted group were fed on days 8 and 10. Six fish from the unfed and refed group were randomly selected whiles four fish were selected from the fed group and the hypothalamus of each fish was collected on specific days, stored at − 80 °C until RNA extraction”.

Q15: Line 156: « … while the other fasted group was » in place of « … whiles the other fasted group were »

Reply: Thank you, it has been replaced.

Q16: Line 157: « … while … » in place of « … whiles »

Reply: Thank you, it has been replaced.

Q17: Line 160: Was the peptide synthesized in amidated form? If yes, please specify.

Reply: Yes, the peptide was synthesized in aminated form. We have clarified this in the revised manuscript.

Q18: Lines 162, 171, 178 : « Npy » in place of « Npy »

Reply: Thank you, it has been replaced.

Q19: Line 194: replace « Where … » by «When… »

Reply: Thank you, it has been replaced.

Q20: Lines 204-207: There is no transmembrane area in the precursor. Please correct the sentence as follows: « It contained a 300-bp ORF that encodes a 99-amino acid (aa) protein (prepro-Npy), which was composed of a signal peptide of 28 aa, a mature 36 aa peptide, 3 aa (GKR) forming the amidation-proteolytic site and a 31 aa-spacer region of the NPY precursor (known as CPON, carboxy terminal peptide of neuropeptide Y in Cerdá-Reverter et al., 2000) (see figure 1a)»

Reply: Thank you, the statement has been rewritten.

Q21: Line 207 « … with the amino acid sequence of spotted scat…»

Reply: This statement has been changed to “A comparison of the spotted scat Npy amino acid sequence was analysed with that of other species”.

Q22: Lines 211-12. There is no transmembrane area in the Npy precursor.

Reply: Thank you, we have crosschecked and made corrections.

Q23: Line 225: A description of the 3D structure of spotted scat Npy is lacking. Please correct.

Reply: Thank you. The authors have agreed to remove all tertiary 3D structures from the manuscript.

Q24: Line 227: Please specify how the percentage similarities between the different Npy sequences is calculated: from the whole precursor or only from the mature peptide?

Reply: The whole precursor was used in calculating the similarity difference.

Q25: Figure 2. Schemes for 3D structures are unreadable because too small. The phylogenetic tree should include more species, in particular non teleosts bony fishes (such as Lepisosteus oculatus) and cartilaginous fishes (e.g. Callorhinchus milii and Squalus acanthias).

Reply: Thank you. The authors have agreed to remove all tertiary 3D structures from the manuscript. Also, we have included more species to the phylogenetic tree.

Q26: Lines 239-41. Reverse transcription-polymerase chain reaction (RT-PCR) is not quantitative method but only quality method. Authors postulate that npy mRNA levels are moderate (line 239) or strong (240) or weak (241). All these words are inappropriate regarded the method used. This study should be performed by using real-time quantitative PCR.

Reply: Thank you, qPCR has been adopted to determine the mRNA level of npy in spotted scat.

Q27: Line 234: The selection of the tissues is not logical. Regarding the central nervous system, there is no reason to restrict the study to the hypothalamus. The other regions of the brain (or at least the whole brain) + the spinal cord should be included in the study.

Reply: Tissues were selected based on previous studies on other fish species. However, we didn’t use the whole brain, other regions or the spinal cord because, we used some parts of the brain for in situ hybridization. Also, reference was given to the hypothalamus because we specifically examined the expression of the gene under different conditions in the subsequent sections.

Q28: Lines 250-51. The choice of the two regions (telencephalon and optic tectum) should be justified.

Reply: Thank you for your suggestion, we have added a description justifying why we selected these two regions of the brain in the discussion section (section 4.3).

Q29: Figure 4. It would have been more appropriate to use colour micrographs to distinguish the labelled cells (hybridization signal) from possible artifacts. The levels of the sections a, b, c and e should be given in the brain scheme (left). All the nuclei visible in the pictures should be named. Pictures d and f are not faithful representations of the boxed regions displayed in pictures c and d, respectively. Please correct. What do the arrows mean in pictures d and f?

Reply: Thank you, the purpose of this particular experiment was to confirm the existence of the gene in the brain of the spotted scat. Thus, much attention was given to the particular sections in these brain regions, subsequent studies will focus more on the specific sections these signals were identified. We have provided a new figure which best describes our experiment. We have described the figure in the results section (section 3.4). Also, we have been able to name the visible nuclei. A description has been provided in the figure legend, indicating what the arrows are for.

Q30: Figure 5 vs Figure 6: Please use the same color code in both figures.

Reply: Thank you. These two figures have the same color code. However, in figure 6, there was another parameter (refed), leading to the addition of another color.

Q31: Lines 284-86: «… although there was a decrease in the expression in the hypothalamus of the unfed group than the fed group». This sentence is not correct. Moreover, it is not clear what the authors are comparing their results to. Please rephrase.

Reply: Thank you for pointing this out. We realized this statement was not needed and has been deleted.

Q32: Line 306: « … while … » in place of « … whiles »

Reply: Thank you, it has been replaced.

Q33: Lines 350-55: « There is a higher level of conservation between spotted scat Npy and other teleost fish species (vertebrates) amino acid sequences … » and « Sequence analysis indicates that spotted scat Npy was highly conserved with the Npy of other vertebrates, particularly, that of other teleost fish species ». These two sentences are redundant. Please correct.

Reply: Thank you, it has been corrected.

Q34: Lines 362-65: Both sentences are redundant. Please rephrase.

Reply: Thank you, corrections have been made.

Q35: Line 363: Please replace « Confirming to studies by …» by « Confirming the studies by …»

Reply: Thank you, it has been corrected.

Q36: Line 391-96. This sentence is too long. Please rephrase.

Reply: Thank you, we have rephrased this statement.

Q37: Line 409: Replace « indicate » by « suggest »

Reply: Thank you, it has been replaced.

Q38: Line 411: « Signals in the optic tectum signify its functions as a feeding center …». This sentence is misleading. Please correct.

Reply: Thank you for the correction. We crosschecked and we have made the necessary corrections here.

Q39: Lines 438-41: This explanation is not convincing. The sequence similarity reported by the authors between Atlantic salmon Npy and spotted scat Npy is due to the fact that only a few species were used for their phylogenetic analysis. Actually, salmoniforms and perciforms are evolutionarily distant species among teleosts.

Reply: Thank you for the input here, this statement has been deleted in the revised manuscript.

Q40: Lines 478, 479, 484, and elsewhere in the manuscript: Please replace «… mRNA expression » by «… mRNA level …».

Reply: Thank you, changes have been made.

Q41: Lines 484-86: This sentence is unclear. What differences are the authors talking about ?

Reply: Thank you. The difference here is between the in vitro ans in vivo effects.

Q42: Line 487: Please replace «… literature …» by «… litterature …»

Reply: Thank you, it has been replaced.

Q43: Line 494: Please replace « These genes » by «These peptides »

Reply: Thank you, it has been replaced.

Q44: Line 497: «… persuaded an increase …». Persuaded in not appropriate in this sentence. Please correct.

Reply: Thank you, it has been corrected.

Q45: Line 515: Please replace « … hasn’t … » by « … has not … »

Reply: Thank you, it has been replaced.

Q46: Line 691: Please replace « Pusetskayaf » by « Plisetskaya ». Make sure that the spelling of all the other authors cited in the paper is correct

Reply: Thank you, it has been replaced.

We have improved the manuscript with reviewer’s good suggestion. Thank you very much for your warm works.

Round 2

Reviewer 2 Report

My comments are attached.

My main concern is about the phylogenetic tree that it has to be rebuilt.  

My other comments relate to the form of the manuscript
